# BALANCING EXTREMES: EXPLOITING THE PERFORMANCE SPECTRUM FROM BEST TO WORST IN MULTI-AGENT SYSTEMS

## ABSTRACT

Coordinating exploration and avoiding suboptimal equilibria remain central challenges in cooperative multi-agent reinforcement learning (MARL). We introduce BEMAS (Balancing Extremes in Multi-Agent Systems), a decentralized and proximity-aware framework that exploits the performance spectrum that naturally emerges as agents learn at different rates. During training, agents exchange bounded local messages to identify their best and worst neighbors via phase-aware TD-error scoring: a curiosity score to encourage coordinated exploration and a performance score to guide exploitation. BEMAS couples two shaped signals: (i) optimism, an intrinsic bonus equal to the optimistic action-value gap, with respect to the best neighbor; and (ii) pessimism, a relative-entropy-based repulsion that discourages imitation of the worst neighbor. A schedule down-weights optimism and up-weights pessimism over training, and execution is fully decentralized with no communication. We establish boundedness of the shaping terms and add a Bayesian stability regularizer that limits policy surprise, resulting in stable updates. Across a standard cooperative MARL benchmark, BEMAS proves superior performance compared to baselines, with ablations isolating the contributions of optimism and pessimism. Motivated by group learning theory, the proposed framework provides a simple mechanism that moves toward the best peers and repels weak behaviors.

## 1 INTRODUCTION

Coordinating exploration while avoiding convergence to poor local equilibria is a central challenge in cooperative multi-agent reinforcement learning, driven by non-stationarity and difficult credit assignment. A key but under-exploited signal is the performance spectrum that naturally emerges as agents learn at different rates, expressing tendencies such as curiosity during early discovery and reliability during consolidation. Motivated by group-learning theory in sociology, where collectives often imitate high performers and avoid behaviors linked to poor outcomes, we hypothesize that agents should learn from both extremes of this spectrum, but in different ways and at different phases: move toward what works best and move away from what works worst.

We instantiate this principle with BEMAS, Balancing Extremes in Multi-Agent Systems, a decentralized, proximity-aware framework that uses only bounded local messaging during training and no communication at execution. Agents rely on their neighborhood and score neighbors by phase: a curiosity score to prioritize informative peers during exploration, and a performance score to identify reliable peers during exploitation. BEMAS then couples two shaping signals, used as reshaped reward terms that augment the environment reward: (i) Optimism as a peer-relative action-value gap bonus that rewards an agent only when its chosen state–action outperforms the best neighbor, and (ii) Pessimism as a behavioral repulsion that penalizes similarity to the worst neighbor, discouraging imitation of weak and suboptimal strategies. A schedule emphasizes optimism early in training and pessimism later.

Because repulsion against a moving reference can destabilize learning, we introduce a Bayesian stability term that penalizes abrupt changes via Bayesian surprise, providing proximal control without suppressing steady improvement. We establish the boundedness of the optimistic and repulsion shaping under support conditions, and prove that the stability penalty admits a bound governed by the

belief step size. These properties ensure well-posed reshaped returns and stable TD learning under a decentralized framework. Empirically, we evaluate on a cooperative predator–prey grid-world, comparing against Independent Q-Learning (IQL) and PED-DQN. Across settings, BEMAS achieves up to $+27.5\%$ higher mean reward than PED-DQN and up to $+132.5\%$ over IQL, while reducing variability with up to $58.5\%$ lower standard deviation vs. IQL and $51.9\%$ vs. PED-DQN. We include ablations that isolate optimism and pessimism, as well as the effect of the stability term.

In summary, our contributions can be summarized as:

- We introduce a *decentralized*, proximity-aware framework that exploits the full performance spectrum; learning from the best while avoiding the worst, using only local messaging during training and no communication at execution.
- We propose a *phase-aware* optimistic gap and KL-based behavioral repulsion.
- We establish key properties: bounded optimistic shaping, convex/bounded KL repulsion with strict separation, and a stable KL-proximal policy update .
- We evaluate on a cooperative MARL benchmark, comparing against IQL and PED-DQN, and include ablations isolating optimism, pessimism, and stability effects.

## 2 RELATED WORK

We situate the proposed approach at the intersection of reward shaping in MARL and optimism/pessimism mechanisms for directed exploration and robustness.

### 2.1 REWARD SHAPING IN MARL

Potential-based reward shaping (PBRS) augments environment rewards with differences of a potential function while preserving optimal policies; later work proved PBRS is equivalent to suitable value initialization and extended it to dynamic settings with guarantees that also apply in multi-agent scenarios (Ng et al., 1999; Wiewiora, 2003; Devlin & Kudenko, 2012). In contrast to shaping-based approaches, early multi-agent methods such as Independent Q-Learning (IQL) treat other agents as part of the environment and learn per-agent value functions (Tan, 1993); while simple and widely used, this perspective highlights the challenges of non-stationarity and credit assignment that reward shaping attempts to mitigate. Unlike PBRS and its dynamic variants, we do not rely on a global potential; instead we shape with *peer-relative* signals, remaining decentralized at execution and proving boundedness of the resulting reshaped returns.

Beyond classic PBRS, multi-agent credit assignment methods such as difference rewards shape the learning signal by measuring each agent's marginal contribution to the team objective, and have been combined with policy gradient methods in cooperative tasks (Castellini et al., 2020). Closer to our interest in socially informed incentives, *peer-based* reshaping has been explored in PED-DQN, where agents exchange peer-evaluation signals to reshape local rewards so that individual updates align better with cooperative outcomes (Hostallero et al., 2020). A complementary direction adds *intrinsic social* signals and maximizes mutual information (MI) to coordinate exploration: agents are rewarded for causal influence on others' actions (Jaques et al., 2019); influence-regularized exploration (EITI) uses information-theoretic penalties (Wang et al., 2020); and VM3-AC maximizes a variational MI bound under CTDE (Kim et al., 2020). Lightweight architectural tweaks can also help in sparse-reward settings (Li et al., 2023).

### 2.2 OPTIMISM AND PESSIMISM IN MARL

**Optimism for Exploration.** Optimism in the face of uncertainty encourages agents to try actions/states that might yield higher returns. Intrinsic-motivation and count-based approaches operationalize this via novelty signals (pseudo-counts) or learning progress, including density-model based pseudo-counts (Bellemare et al., 2016; Ostrovski et al., 2017; Tang et al., 2017), information-gain/curiosity methods (Houthooft et al., 2016; Burda et al., 2018; Pathak et al., 2019), and the random-network-distillation (RND) bonus (Burda et al., 2019). In actor–critic RL, *Optimistic Actor–Critic* (OAC) shows that maintaining optimistic value bounds can improve directed exploration without destabilizing learning (Ciosek et al., 2019). In cooperative MARL, optimism has also

been encoded via asymmetric updates: hysteretic Q-learning down-weights negative TD errors to stabilize coordination, while lenient learners apply decaying "leniency" to early negative feedback so agents begin optimistic and gradually anneal (Matignon et al., 2007; Palmer et al., 2018). Weighted QMIX further introduces an *optimistically weighted* variant that emphasizes higher-value joint actions during value factorization (Rashid et al., 2020). The proposed optimism is neither count-based nor bound-propagation based; it is a *local advantage-like discrepancy* between the agent and its best neighbor, shaped via an action-value gap to encourage exploration.

**Pessimism.** On the other side of the spectrum, pessimism is widely used to control over-estimation and improve stability. In continuous control, TD3 forms conservative targets via the minimum of two critics ;clipped double Q (Fujimoto et al., 2018), and Maxmin Q-learning extends this with ensembles to control estimation bias (Lan et al., 2020). In offline RL, CQL regularizes Q-values to discourage out-of-distribution actions, and its penalty is often adapted to online settings (Kumar et al., 2020). Bridging stances, Tactical Optimism–Pessimism switches between optimistic and pessimistic quantile critics via a bandit (Moskovitz et al., 2022). Unlike conservative targets, we impose *policy-level* pessimism via Kullback-Leiber divergence to actively discourage imitation of weak peers.

**Combining Optimism and Pessimism.** Yang et al. (2024) explicitly decouple exploration and utilization with dual actors—an optimistic actor that explores using overestimated values and a pessimistic actor that selects safer, underestimated values—showing that alternating/scheduling these roles can retain exploration while stabilizing policy improvement. Related ideas conditionally introduce optimism in cooperative MARL to drive exploration when it is likely to be beneficial (Zhao et al., 2023). BEMAS combines both signals in a single reshaped reward with a simple phase schedule; unlike dual-actor CTDE schemes, the proposed approach uses only local messaging during training.

## 3 PRELIMINARIES

We model the cooperative multi-agent control problem as a Dec-POMDP:

$$\mathcal{M} = \langle \mathcal{V}, \mathcal{S}, \{\mathcal{U}_i\}_{i \in \mathcal{V}}, \{\mathbf{O}_i\}_{i \in \mathcal{V}}, P, O, r, \gamma \rangle,$$

where $\mathcal{V} = \{1, \ldots, \nu\}$ is the set of agents, $\mathcal{S}$ the state space, $\mathcal{U}_i$ and $\mathbf{O}_i$ are the action and observation spaces of agent $i$, $P(s' \mid s, u_{1:\nu})$ is the transition kernel, and $O$ the observation kernel (commonly factorized as $O(o'_{1:\nu} \mid s', u_{1:\nu}) = \prod_{i=1}^{\nu} O_i(o'_i \mid s'))$. At time $t$, the environment is in state $s^t \in \mathcal{S}$, each agent $i$ receives a local observation $o_i^t \sim O_i(\cdot \mid s^t)$, selects action $u_i^t \sim \pi_i(\cdot \mid h_i^t)$ from a stochastic policy $\pi_i : \mathcal{H}_i \to \Omega(\mathcal{U}_i)$ based on its local history $h_i^t = (o_i^0, u_i^0, \ldots, o_i^t)$, and the system transitions to $s^{t+1} \sim P(\cdot \mid s^t, u_{1:\nu}^t)$ with a shared team reward $r^t = r(s^t, u_{1:\nu}^t, s^{t+1}) \in \mathbb{R}$. We consider the discount factor $\gamma \in (0, 1)$. The joint policy factorizes as $\pi(u_{1:\nu} \mid h_{1:\nu}) = \prod_{i=1}^{\Omega} \pi_i(u_i \mid h_i)$.

Both training and execution are decentralized: there is no access to joint states/actions and no global replay. Each agent maintains its own parameters $\theta_i$ and updates them from locally available trajectories (and, when permitted by the environment, uses peer statistics defined in Section 4) without centralized supervision. At execution time, each agent acts using only its own history $h_i^t$.

Agents are embedded in a metric workspace $(\mathcal{X}, d)$ through a state-dependent placement map $\Phi : \mathcal{S} \to \mathcal{X}^{\nu}$, so that $\Phi(s) = (x_1, \ldots, x_{\nu})$ gives the coordinates of agents in state $s$. For a communication radius $\rho > 0$, the *proximity-based neighborhood* of agent $i$ at time $t$ is:

$$\mathcal{N}_i^t = \{ j \in \mathcal{N} : d(x_i^t, x_j^t) \leq \rho \} \cup \{i\}, \qquad x_{1:\nu}^t = \Phi(s^t). \tag{1}$$

All communication is local: when messaging is allowed during *training*, agent $i$ may receive a bounded-size signal from agents in $\mathcal{N}_i^t$; at *execution*, policies depend only on $h_i^t$.

## 4 METHODOLOGY

We propose to exploit information at *both extremes* of the performance spectrum in multi-agent learning: agents should *learn from the best* (to accelerate exploration and propagate successful behaviors) while simultaneously *avoiding the worst* (to reduce the chance of imitating sub-optimal strategies). The proposed design remains fully decentralized at execution with no communication and relies only on local interactions during training.

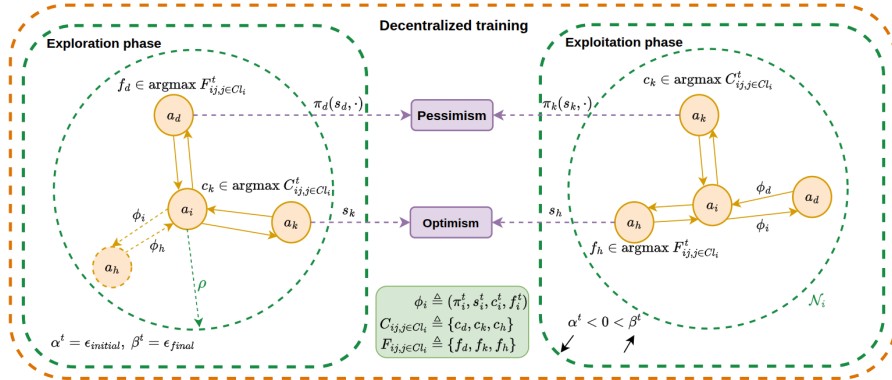

Figure 1: Each agent $i$ rely on its neighborhood $\mathcal{N}_i$ for local messaging; in exploration, neighbors are ranked by $c^t$ and an optimism bonus is applied, while in exploitation they are ranked by $f^t$ and a KL repulsion from the worst neighbor is added.

**Phase-based neighbor scoring.** For each neighbor $j \in \mathcal{N}_i^t$, we define a phase-dependent score:

$$\eta^t(j) = \begin{cases} c^t(j), & \text{exploration (curiosity)} \\ f^t(j), & \text{exploitation (performance)} \end{cases} \tag{2}$$

**Curiosity score.** For agent $j$, we use a TD-error magnitude as a decentralized curiosity score:

$$\delta_j^t = r^t + \gamma \max_{a'} Q_j(o_j^{t+1}, a') - Q_j(o_j^t, a_j^t), \qquad c^t(j) = (1-\lambda)\, c^{t-1}(j) + \lambda\, |\delta_j^t|, \tag{3}$$

with the exponential moving average (EMA) rate $\lambda \in (0,1)$.

**Performance score.** During exploitation we favor agents with *small prediction error*. We define [1]:

$$f^t(j) = \frac{1}{(1-\lambda)\, f^{t-1}(j) + \lambda\, |\delta_j^t|}, \tag{4}$$

During *exploration*, we instantiate $c_i^t$ as a curiosity score given by the running magnitude of the TD error; agents with larger $c_i^t$ (i.e., larger $|\delta_i^t|$) are regarded as more curious and thus ranked as the "best" for this phase. Conversely, during *exploitation* we use a performance score $f_i^t$ derived from the same signal but with the opposite preference: agents with *smaller* prediction error (smaller $|\delta_i^t|$) are deemed more reliable and are therefore ranked as the "best" in this phase. We select the *best* and *worst* neighbors (respectively) as: $b_i^t \in \arg\max_{j \in \mathcal{N}_i^t} \eta^t(j)$ and $w_i^t \in \arg\min_{j \in \mathcal{N}_i^t} \eta^t(j)$.

**Communication between agents.** Per step and per agent, the following are exchanged *within* $\mathcal{N}_i^t$ during training: (i) scalar scores ($c^t(\cdot)$ and $f^t(\cdot)$) for neighbor selection; (ii) a representation of the current policy to compute KL terms; and (iii) the Q-values of the current state and chosen action of every agent. Now that the best and worst neighbors are defined, we introduce the proposed pessimism and optimism reshaping mechanisms.

### 4.1 ACTION-VALUE GAP FOR OPTIMISTIC SHAPING

We view optimism through a *discrepancy* lens: the quality of agent $i$'s chosen state–action pair is judged relative to the best local peer in its neighborhood $\mathcal{N}_i$. Like an advantage function, the proposed gap is a discrepancy between two $Q$-values: the agent's estimate for $(s_i^t, u_i^t)$ measured against a peer's $Q$ taken as the reference. The latter is selected by the curiosity-based score $\eta^t$ during exploration (see Fig.1). We grant a bonus only when this discrepancy is positive; when agent $i$ outperforms the most exploratory peer, which implements optimism and encourages exploration.

---

[1]Empirically, $f^t(j) = 1/(\text{EMA}(|\delta_j|))$ is more stable than $1/|\delta_j^t|$ and aligns with the intuition that low TD error correlates with higher performance late in training.

**Definition 4.1 (Action–value gap)** *Fix agent $i$ at time $t$. Let $\mathcal{S} \subseteq \mathbb{R}^d$ be the (joint) state space with $d = \dim(\mathcal{S})$, and let $b_i^t$ denote $i$'s best neighbor. We define the optimistic state–action gap as:*

$$\Delta_i^t := \left[ Q_i(s_i^t, u_i^t) - Q_{b_i}(s_{b_i}^t, u_{b_i}^t) \right]_+, \qquad [x]_+ := \max\{x, 0\}, \quad Q : \mathbb{R}^d \times \mathcal{U} \to \mathbb{R}. \quad (5)$$

**Optimism as an intrinsic reward.** We shape rewards with a monotone transform of the positive gap:

$$\Psi_i^t = \Psi_i^t(s_i^t, u_i^t; b_i^t) = \psi_{\mathrm{opt}} \, g\left(\Delta_i^t\right) \mathbf{1}\{\Delta_i^t > 0\}, \qquad g(x) = log(1 + x) \quad (6)$$

with $\psi_{opt}$ being the optimism parameter used. When $\Delta_i^t = 0$, the agent does not get a bonus $\Psi_i^t = 0$. Otherwise, the agent is rewarded in proportion to the margin by which it outperforms its best neighbor. Using a logarithmic transform, $\Psi_i^t = \psi_{\mathrm{opt}} \log(1 + \Delta_i^t)$ is monotone and concave, resulting in bounded gradients, and robustness to reward rescaling and outliers (see Lemma B.1 in App. B.1).

**Theorem 4.2 (Boundedness of the optimistic intrinsic reward)** *The environment reward is bounded $|r^t| \leq r_{\max}$ and $\gamma \in (0, 1)$. This implies that any agent's optimal action-value is bounded: $|Q_i(\cdot, \cdot)| \leq Q_{\max} := \frac{r_{\max}}{1-\gamma}$, and likewise for $Q_{b_i}$. Therefore:*

$$0 \leq \left| Q_{b_i}(s_{b_i}^t, u_{b_i}^t) \right| + \left| Q_i(s_i^t, u_i^t) \right| \leq 2Q_{\max}, \Rightarrow \quad 0 \leq \Psi_i^t \leq \psi_{\mathrm{opt}} \, log\left(1 + 2Q_{\max}\right)$$

Full proof of this inequality can be found in App.B.2.

**Optimistic shaping schedule.** We use a shaping schedule to induce optimism during the exploration phase. Let $T_{\mathrm{exp}}$ be the exploration horizon. A simple piecewise-linear schedule is:

$$\alpha^t = \begin{cases} \alpha_{\min} + (\alpha_{\max} - \alpha_{\min}) \dfrac{T_{\mathrm{exp}}}{T_{\mathrm{exp}} - t}, & t < T_{\mathrm{exp}}, \\ \alpha_{\min}, & t \geq T_{\mathrm{exp}}, \end{cases} \quad \text{clipped to } [\alpha_{\min}, \alpha_{\max}]. \quad (7)$$

### 4.2 BEHAVIORAL REPULSION VIA RELATIVE ENTROPY

Inspired by fear and approval dynamics in group learning, we discourage agent $i$ from imitating weak local neighbors: during *exploration*, it steers away from the least curious neighbor, and during *exploitation*, from the least performing one.

We induce pessimism as *behavioral repulsion* from agent $i$'s worst neighbor $w_i^t$ using the Kullback–Leibler (KL) divergence. This relative entropy is the expected log-likelihood ratio of one distribution *relative* to another. Let $\pi_i(\cdot \mid s)$ be agent $i$'s policy and $\pi_{w_i^t}(\cdot \mid s)$ that of its worst neighbor, the per-state KL is:

$$D_{\mathrm{KL}}\left(\pi_i(\cdot \mid s_i^t) \,\|\, \pi_{w_i^t}(\cdot \mid s_{w_i^t}^t)\right) = \sum_{u \in \mathcal{U}} \pi_i(u \mid s_i^t) \, \log \frac{\pi_i(u \mid s_i^t)}{\pi_{w_i^t}(u \mid s_{w_i^t}^t)} . \quad (8)$$

This distance is equal to $0$ if and only if the two policies coincide, and increases as $\pi_i$ departs from $\pi_{w_i^t}$, making it a simple repulsive regularizer. The pessimistic shaping term is defined per step as:

$$\Gamma_i^t = \Gamma_i^t(\pi_i^t, \pi_{w_i}^t) = D_{\mathrm{KL}}\left(\pi_i^t(\cdot \mid s_i^t) \,\|\, \pi_{w_i^t}^t(\cdot \mid s_{w_i^t}^t)\right) \quad (9)$$

The further away an agent $i$'s policy is from its worst peer's $w_i$, the higher its intrinsic reward.

**Pessimistic shaping schedule.** We smoothly increase pessimism from exploration to exploitation using a piecewise-linear schedule, with $T_{\max}$ being the total number of time-steps during training:

$$\beta^t = \begin{cases} \beta_{\min}, & t < T_{\mathrm{exp}}, \\ \beta_{\min} + (\beta_{\max} - \beta_{\min}) \dfrac{t - T_{\mathrm{exp}}}{T_{\max} - T_{\mathrm{exp}}}, & t \geq T_{\mathrm{exp}}, \end{cases} \quad \text{clipped to } [\beta_{\min}, \beta_{\max}]. \quad (10)$$

Figure 2: Overview of the BEMAS architecture. Agents interact with the environment by selecting actions according to their policies. The best and worst neighbors are identified, and the reshaped reward is computed from communicated signals. The purple circles denote the three mechanisms: Bayesian stability, behavioral repulsion, and optimistic action-value gap.

**Theorem 4.3 (Upper bound of the repulsion)** *For fixed $\pi_{w_i^t}(\cdot|s)$, for state $s$, we define:*

$$p_n^t(s) \; := \; \max_{u \in U:\, \pi_{w_i^t}^t(u|s) > 0} \left| \frac{\pi_i^t(u \mid s)}{\pi_{w_i^t}^t(u \mid s)} - 1 \right|. \tag{11}$$

*and we get for any state distribution $\mu$:*

$$\mathbb{E}_{s \sim \mu}\Big[ D_{\mathrm{KL}}\big(\pi_i^t \| \pi_{w_i^t}^t\big) \Big] \; \leq \; \mathbb{E}_{s \sim \mu}\Big[ \big(p_n^t(s)\big)^2 \Big] \; \leq \; c_1 \cdot \big(p_n^t\big)^2 \tag{12}$$

*with $\mathbb{E}_{s \sim \mu}\Big[ \big(p_n^t(s)\big)^2 \Big] \; = \; \mathcal{O}\Big( \big(p_n^t\big)^2 \Big)$ and $c_1$ is a positive constant.*

Proof of this theorem can be found in App.B.3.

**Theorem 4.4 (Lower bound of the repulsion (Gibbs' inequality))** *Let $\pi_i(\cdot \mid s_i^t)$ and $\pi_{w_i^t}(\cdot \mid s_{w_i^t}^t)$ be action distributions over a finite action set $\mathcal{U}$, with $\pi_{w_i^t}(u \mid s_{w_i^t}^t) > 0$ for all $u \in \mathcal{U}$. Then:*

$$D_{\mathrm{KL}}\big(\pi_i(\cdot \mid s_i^t) \, \| \, \pi_{w_i^t}(\cdot \mid s_{w_i^t}^t)\big) \; \geq \; 0, \tag{13}$$

From equations 12 and 13, we get the bounds of the KL-based shaping term for behavioral repulsion:

$$0 \leq \beta^t \Gamma_i^t(\pi_i^t, \pi_{w_i}^t) \leq c_1 \cdot \big(p_n^t\big)^2 \tag{14}$$

**Assumption 4.5 (Instability risk)** *With a non-stationary reference $\pi_{w_i^t}$, the repulsion bonus $\beta^t \Gamma_i^t$, if used alone, can induce unstable learning or divergent updates when the reference changes rapidly.*

Following this assumption, we introduce a Bayesian inference-based intrinsic reward to induce more stability in the policies' updates.

## 4.3 Stability via a Bayesian policy belief

The proposed repulsion term can destabilize learning (see Assumption 4.5): if neighbors keep changing, penalizing divergence may make agents chase moving targets. We add a *Bayesian stability* bonus that discourages sudden shifts in an agent's own policy while still letting it move away from bad behavior. The intuition is simple: maintain a belief over the policy and penalize the information change, hence Bayesian surprise, in that belief.

**Bayesian policy belief and update.** Let $\mathcal{U}$ be a finite action set of size $K$. For each agent $i$, we maintain a Dirichlet belief over action probabilities with parameters $\boldsymbol{\xi}_i^t \in \mathbb{R}_{>0}^K$, total mass $\xi_{i,0}^t = \sum_{u \in \mathcal{U}} \xi_{i,u}^t$, and mean $\tilde{\pi}_i^t = \boldsymbol{\xi}_i^t / \xi_{i,0}^t$. We apply discounted soft-count updates:

$$\boldsymbol{\xi}_i^t \;\leftarrow\; \varphi\,\boldsymbol{\xi}_i^{t-1} + \omega^t\,\pi_i^t, \qquad \tilde{\pi}_i^t \;=\; (1-\tau^t)\,\tilde{\pi}_i^{t-1} + \tau^t\,\pi_i^t, \qquad \tau^t \;=\; \frac{\omega^t}{\varphi\,\xi_{i,0}^{t-1} + \omega^t} \in (0,1). \quad (15)$$

with $\varphi \in (0,1)$ being the forgetting factor that exponentially downweights past evidence, and $\omega^t > 0$ being the evidence weight that adds the current evidence. Normalizing gives $\tilde{\pi}_i^t$, whose update in equation 15 is an EMA with data-dependent step size $\tau^t \in (0,1)$. It is a combination of the previous belief mean and the current policy with step size $\tau^t$ that is reduced as the effective sample size $\xi_{i,0}^{t-1}$ increases. Updates become smaller, and the belief stabilizes while still adapting when $\varphi < 1$.

**Bayesian surprise and stability bonus.** We measure belief change via the categorical KL between successive means:

$$\Lambda_i^t \;=\; D_{\mathrm{KL}}\big(\tilde{\pi}_i^{t-1} \,\big\|\, \tilde{\pi}_i^t\big), \tag{16}$$

and add $-\Lambda_i^t$ to the reward, which allows penalizing sudden policy shifts while allowing improvement.

**Theorem 4.6 (Boundedness of Bayesian surprise)** *For state $s$, let $P = \tilde{\pi}_i^{t-1}(\cdot \mid s)$ and $R = \pi_i^t(\cdot \mid s)$ denote the Dirichlet beliefs at different time-steps. We assume $P(u) > 0$ for all available actions $u \in \mathcal{U}$ and set a constant $c_2 > 0$. We define:*

$$q^t(s) \;:=\; \max_{u:\, P(u)>0} \left| \frac{R(u)}{P(u)} - 1 \right| \tag{17}$$

*Then the Bayesian surprise satisfies:*

$$D_{\mathrm{KL}}\big(P \,\|\, (1-\tau^t)P + \tau^t R\big) \;=\; \mathcal{O}\!\left( \tfrac{(\tau^t)^2}{1-\tau^t}\big(q^t(s)\big)^2 \right) \;\leq\; c_2 \cdot \tfrac{(\tau^t)^2}{1-\tau^t}\big(q^t(s)\big)^2 \tag{18}$$

Proof of this theorem can be found in App.B.3.

## 4.4 OVERALL FRAMEWORK

We train $Q_i$ with standard TD updates using a reshaped reward $\tilde{r}_i^t$:

$$y_i^t = \tilde{r}_i^t + \gamma \max_{u'} \bar{Q}_i(s_i^{t+1}, u'), \qquad \mathcal{L}_{Q_i} \;=\; \big(y_i^t - Q_i(s_i^t, u_i^t)\big)^2, \tag{19}$$

where $\bar{Q}_i$ is a target network (see Fig. 2). The per-step reshaped reward for agent $i$ is:

$$\tilde{r}_i^t \;=\; r^t \;+\; \alpha^t \Psi_{i,\mathrm{opt}}^t \;+\; \beta^t \Gamma_i^t \;-\; \Lambda_i^t \tag{20}$$

where $\alpha^t \geq 0$ modulates optimism and $\beta^t \geq 0$ modulates pessimism.

**Assumption 4.7 (Modified Dec-POMDP)** *Each agent $i$ receives the reshaped reward $\tilde{r}_i^t$. Each term of this reward is measurable w.r.t. the local history and in-neighborhood messages $\{\mathbf{m}_{j \to i}^t\}_{j \in \mathcal{N}_i^t}$. Training with $\tilde{r}$ is thus equivalent to solving the modified Dec-POMDP:*

$$\tilde{\mathcal{M}} = \langle \mathcal{V}, \mathcal{S}, \{\mathcal{U}_i\}_i, \{\mathcal{O}_i\}_i, P, O, \tilde{r}, \gamma \rangle.$$

We study the boundedness of each shaping term to ensure $\tilde{r}$ is uniformly bounded and discounted returns are finite. Using 7, 12 and 18, we get for all $i, t$:

$$\big|\tilde{r}_i^t\big| \;\leq\; r_{\max} + \alpha_{\max}\,\psi_{\mathrm{opt}} \log\!\big(1 + 2Q_{\max}\big) + \beta_{\max}\,c_1\big(p_n^t\big)^2 + c_2 \cdot \tfrac{(\tau)t^2}{1-\tau^t}\big(q^t(s)\big)^2, \tag{21}$$

So $\tilde{\mathcal{M}}$ is well-posed and admits optimal joint policies under standard Dec-POMDP assumptions.

## 5 EXPERIMENTS

**Baselines.** Since the proposed method relies on *reward reshaping* in a decentralized framework, we compare against two baselines: (i) **IQL**, a decentralized, no-shaping baseline; and (ii) **PED-DQN**, a reward-reshaping DQN designed to promote cooperation in multi-agent settings.

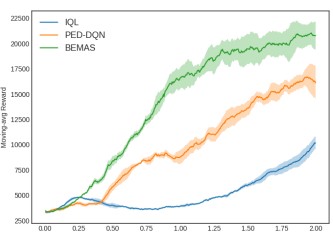 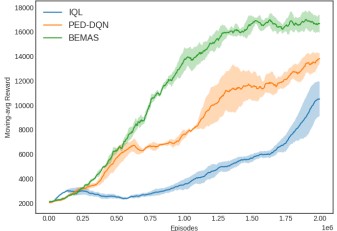 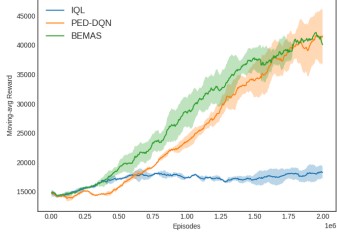

(a) Performance of 20 predators and 19 preys in a 12 by 12 grid.

(b) Performance of 20 predators and 19 preys in a 13 by 13 grid.

(c) Performance of 26 predators and 25 preys in a 12 by 12 grid.

Figure 4: Learning performance of BEMAS compared to IQL and PED-DQN across different settings.

**Environment Details.** We evaluate in a discrete grid-world that models predator–prey interactions and incentive mechanisms. The grid comprises empty cells and walls; a fixed population of predators and preys occupy distinct cells and move in the four cardinal directions under boundary and collision constraints. Predators act under partial observability. Prey attempt to evade capture. A prey is captured if and only if it is fully surrounded by predators (see Fig. 3).

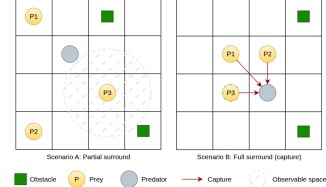

Figure 3: Prey–predator environment.

### 5.1 RESULTS

Across all three scenarios Fig. 4, BEMAS outperforms both baselines in terms of sample efficiency, final reward, and stability. In Figs. 4a and 4b, BEMAS begins to accelerate after only $\sim$0.25M episodes. In Setting (a), BEMAS reaches a reward of 20,697$\pm$1,381, compared to 16,239$\pm$1,075 for PED-DQN and 8,904$\pm$902 for IQL, corresponding to relative gains of **+27.5%** and **+132.5%** (see App. 2). The optimistic shaping signal induces early exploration, avoiding the stagnation seen in IQL. PED-DQN benefits from peer-evaluation rewards, but its returns remain substantially below BEMAS.

**Early phase (fast lift, higher variance).** During the first $\sim$0.2–0.6M episodes, BEMAS rises sharply but shows larger variance than PED-DQN. This behavior is expected: optimism dominates the shaping schedule, so agents preferentially learn from the most exploratory peers, producing both rapid learning and greater spread across seeds. Quantitatively, in Setting (b), BEMAS already surpasses PED-DQN by **+26.9%** in mean return, 16,737 vs. 13,192, while maintaining a lower coefficient of variation. IQL, lacking any shaping, improves slowly and saturates at much lower values.

**Late phase (variance contracts).** As training proceeds, weight shifts from optimism to pessimism, introducing repulsion from the worst-performing neighbors and reducing imitation of poor local minima. In Setting (a), BEMAS matches PED-DQN in variability while exceeding it in median reward **+22.3%**. In Setting (c), BEMAS achieves 37,126$\pm$2,006 as a return, slightly below PED-DQN 40,473$\pm$4,169; **-8.3**%, but with lower standard deviation (std): reduced by **51.9%**. This demonstrates that the pessimism phase effectively stabilizes performance across seeds.

**Scaling with agent density.** The densest scenario, in Fig. 4c, is substantially more challenging due to increased agent interactions. Here, BEMAS initially leads with strong improvements and maintains its robustness throughout training, while PED-DQN eventually catches up in mean return but at the cost of far greater variability. Relative to IQL, BEMAS more than doubles the mean reward **+106.3%**, and reduces variance by nearly half. This robustness under higher density validates the phase-scheduled, two-sided shaping that results in both faster and more reliable convergence.

### 5.2 ABLATION STUDY

We isolate the two shaping components of BEMAS: (i) **OMAS** (Optimism in Multi-Agent Systems): we keep only the optimistic action-value gap with the stability regularizer. This tests whether optimism alone can drive coordinated exploration and propagate promising behaviors early in training. (ii)

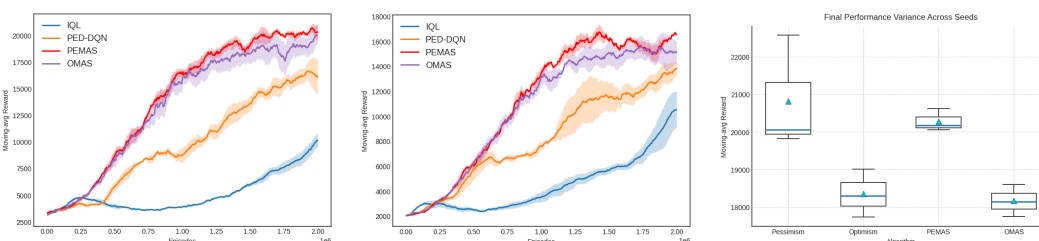

(a) Performance of 20 predators and 19 preys in a 12 by 12 grid.

(b) Performance of 20 predators and 19 preys in a 13 by 13 grid.

(c) Variance study across multiple seeds for ablations.

Figure 5: Ablation study of BEMAS components: optimism, pessimism, and their stabilized variants.

**PEMAS** (Pessimism in Multi-Agent Systems): we keep only the KL-based repulsion from the worst neighbor with the stability regularizer. This evaluates how much of BEMAS's final performance comes from discouraging imitation of persistently weak local policies.

**Exploration dynamics and final performance.** OMAS exhibits higher variance across seeds during the exploration phase. This is consistent with agents receiving positive intrinsic bonuses whenever they outperform the current phase leader, encouraging broader search and more diverse exploration trajectories. As shown in Fig 5a and Fig 5b, PEMAS consistently surpasses OMAS and matches or approaches the full model late in training. Penalizing similarity to the worst neighbor steers policies away from sub-optimal local equilibria, improving exploitation stability and raising asymptotic returns; whereas, optimism's main benefit is earlier, more adventurous exploration.

**Stability and robustness.** As shown in Fig. 5c, the pessimism-only variant reaches the highest reward; max **22,268**, mean **20,805**, but exhibits high variability; std of **1,262**. Incorporating stability achieves comparable performance; mean reward **20,287**, with reduced variance: **430**, lowest overall. In contrast, the optimism-only variant results in lower returns; mean **18,957**, std **396**, and the stabilized optimism variant remains similarly limited; mean **18,148**, std **485** (see App. 3). These results indicate that $\Lambda$ is most effective when combined with repulsion, while offering modest benefits for optimism.

# 6 CONCLUSION AND FUTURE WORK

We introduced BEMAS, a decentralized, proximity-aware shaping framework that exploits both extremes of agents performance. The optimism signal $\Psi_i^t$, a peer-relative action-value gap, rewards agent $i$ only when its Q-value exceeds the best neighbor $b_i^t$, using a concave transform for bounded, well-scaled bonuses; the pessimism signal $\Gamma_i^t$ repels imitation of the weakest neighbor $w_i^t$. These signals are phase-scheduled; curiosity vs. performance leaders, and complemented by a *Bayesian stability* penalty via Dirichlet updates to reduce sudden policy shifts.

The shaping is monotone and bounded under standard $Q$ limits; the repulsion admits a quadratic control bound under absolute continuity; and the stability penalty has an explicit step-size bound, ensuring well-posed discounted returns. Empirically, in cooperative predator–prey setting, BEMAS outperforms baselines IQL and PED-DQN, with ablations isolating optimism, pessimism, and stability effects. Training requires only local messaging and no execution-time communication, preserving decentralization while encouraging exploration and avoiding suboptimal equilibria.

**Limitations and Future Work.** BEMAS provides a theoretically grounded shaping framework with encouraging empirical results, but limitations remain. The proposed guarantees rely on assumptions that suit some settings; for example, with bounded $Q$ values and reliable local messaging, but may not extend to others. In dynamic or resource-constrained deployments, communication assumptions may fail, calling for more adaptable support conditions. Future directions include: (i) extending evaluation to a wider suite of MARL benchmarks to test scalability and generalization. (ii) developing adaptive schedules for optimism and pessimism via task-dependent weighting rather than fixed phases. (iii) relaxing the local messaging assumptions to better align with real-world multi-agent systems.

## 7 REPRODUCIBILITY STATEMENT

We aim for full reproducibility and provide all components needed to reproduce the results reported in this work. The code can be found in this anonymous GitHub repository `https://github.com/Epsilon314159/BEMAS`. Algorithmic details are specified in Algorithm 1, and hyperparameters shared across methods are listed in Table D.3. The partial observability, capture mechanics, and grid details appear in the Environment Details Section 5 and in the map files in the code. Proofs for the robustness of logarithmic shaping, boundedness of the optimistic bonus, and the upper bound for the KL repulsion are included in App.B with explicit conditions and intermediate steps.

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

## A    TABLE OF NOTATION

Below is a table with the mathematical notations that were used in this work.

Table 1: Mathematical notation used in BEMAS.

| Symbol | Description |
|---|---|
| $\mathcal{V}$ | Set of agents (indices $1{:}n$). |
| $\mathcal{N}_i^t$ | Proximity (radius-$\rho$) neighborhood of agent $i$ at time $t$ (time-varying local ensemble). |
| $b_i^t,\ w_i^t$ | Indices of the *best* and *worst* neighbor of agent $i$ under the phase score $\eta^t(\cdot)$. |
| $\eta^t(j)$ | Phase-based neighbor score; $\eta^t(j) = c^t(j)$ in exploration, $\eta^t(j) = f^t(j)$ in exploitation. |
| $c^t(j)$ | Curiosity score of $j$: EMA of TD-error magnitude. |
| $f^t(j)$ | Reliability score of $j$: inverse-EMA of TD-error magnitude. |
| $\Delta_i^t$ | Peer-relative *positive* $Q$-gap: $\left[\, Q_i(s_i^t, u_i^t) - Q_{b_i^t}(s_{b_i^t}^t, u_{b_i^t}^t)\, \right]_+$. |
| $\Psi_i^t$ | Optimistic bonus: $\psi_{\mathrm{opt}} \log\!\big(1 + \Delta_i^t\big)$. |
| $\Gamma_i^t$ | Pessimism KL-based repulsion: $D_{\mathrm{KL}}\!\big(\pi_i^t(\cdot\,|\,s_i^t)\,\big\|\,\pi_{w_i^t}^t(\cdot\,|\,s_{w_i^t}^t)\big)$. |
| $\boldsymbol{\xi}_i^t$ | Dirichlet belief parameters over $\pi_i$ (per-agent action-probability ensemble). |
| $\tilde{\pi}_i^t$ | Dirichlet mean: $\tilde{\pi}_i^t = \boldsymbol{\xi}_i^t/(\mathbf{1}^\top\boldsymbol{\xi}_i^t)$. |
| $\Lambda_i^t$ | Bayesian surprise (stability penalty): $D_{\mathrm{KL}}(\tilde{\pi}_i^{t-1}\,\|\,\tilde{\pi}_i^t)$. |
| $\tau^t$ | Belief step-size: $\tau^t = \dfrac{\omega^t}{\varphi\,\xi_{i,0}^{t-1} + \omega^t} \in (0,1)$. |
| $\psi_{\mathrm{opt}}$ | Optimism scale used in the action-value gap bonus. |
| $\alpha^t, \beta^t$ | Weights for optimism $\Psi$ and pessimism $\Gamma$ in the reshaped reward. |
| $Q_{\max}$ | Uniform $Q$ bound: $Q_{\max} = r_{\max}/(1-\gamma)$. |
| $\mathcal{B}$ | Replay buffer used for off-policy updates. |

## B    MATHEMATICAL PROOF

### B.1    ROBUSTNESS OF LOGARITHMIC SHAPING

**Lemma B.1 (Robustness to rescaling and outliers)** *Let*

$$\Psi(\Delta) = \psi_{\mathrm{opt}} \log(1 + \Delta), \qquad \Delta \geq 0,\ \psi_{\mathrm{opt}} > 0$$

*Then $\Psi$ is nonnegative, strictly increasing, and concave with globally bounded slope; it is stable under reward rescaling, and large outliers only change it sublinearly:*

$$\Psi'(\Delta) = \frac{\psi_{\mathrm{opt}}}{1 + \Delta} > 0, \qquad \Psi''(\Delta) = -\frac{\psi_{\mathrm{opt}}}{(1 + \Delta)^2} < 0, \qquad |\Psi'(\Delta)| \leq \psi_{\mathrm{opt}},$$

*and for any $c > 0$ and $\alpha \geq 1$,*

$$\big|\Psi(c\Delta) - \Psi(\Delta)\big| \leq \psi_{\mathrm{opt}}\,|\log c|, \qquad \Psi(\alpha\Delta) - \Psi(\Delta) \leq \psi_{\mathrm{opt}} \log \alpha$$

**Proof**    The derivative and curvature are:

$$\Psi'(\Delta) = \frac{\psi_{\mathrm{opt}}}{1 + \Delta} > 0, \qquad \Psi''(\Delta) = -\frac{\psi_{\mathrm{opt}}}{(1 + \Delta)^2} < 0$$

Therefore, increases in $\Delta$ always result in a positive bonus (monotonicity), but with diminishing returns (concavity).

Because $1 + \Delta \geq 1$ and $\log(1 + \Delta) \geq 0$ for $\Delta \geq 0$, the slope is uniformly bounded as $|\Psi'(\Delta)| \leq \psi_{\mathrm{opt}}$ and cannot introduce arbitrarily large gradients.

**Robustness to rescaling.**    We consider multiplying all gaps by a constant $c > 0$. We denote $x = \Delta \geq 0$:

$$\Psi(c\Delta) - \Psi(\Delta) = \psi_{\mathrm{opt}}[\log(1 + cx) - \log(1 + x)] = \psi_{\mathrm{opt}} \log\frac{1 + cx}{1 + x}$$

If $c \geq 1$ then $1 + cx \leq c(1 + x)$, hence:

$$0 \leq \log\frac{1 + cx}{1 + x} \leq \log c$$

If $0 < c \leq 1$ the inequalities reverse and we get:

$$\log c \leq \log\frac{1 + cx}{1 + x} \leq 0$$

In both cases, we have:

$$\left|\Psi(c\Delta) - \Psi(\Delta)\right| \leq \psi_{\text{opt}} \left|\log c\right|$$

Hence, using this optimistic shaping term, rescaling rewards cannot distort the shaping by more than a constant shift proportional to $|\log c|$.

**Outlier suppression.** For a multiplicative amplification $\alpha \geq 1$ of a single gap, we have:

$$1 + \alpha x \leq \alpha(1 + x) \quad \Rightarrow \quad \Psi(\alpha\Delta) - \Psi(\Delta) = \psi_{\text{opt}} \log\frac{1 + \alpha x}{1 + x}$$

$$\leq \psi_{\text{opt}} \log\alpha$$

Therefore, extreme gaps only increase the shaping logarithmically, preventing a few large values from dominating the shaping signal. This sublinear growth represents the robustness property of $\log(1 + \Delta)$.

## B.2 Boundedness and Convexity of the Optimistic Intrinsic Reward

**Theorem B.2** *The environment reward is bounded $|r^t| \leq r_{\max}$ and $\gamma \in (0,1)$. This implies that any agent's optimal action-value is bounded by:*

$$|Q_i(\cdot,\cdot)| \leq Q_{\max} := \tfrac{r_{\max}}{1-\gamma}$$

*and likewise for $Q_{b_i}$. Therefore:*

$$0 \leq \left|Q_{b_i}(s_{b_i}^t, u_{b_i}^t)\right| + \left|Q_i(s_i^t, u_i^t)\right| \leq 2Q_{\max}, \quad \Rightarrow \quad 0 \leq \Psi_i^t \leq \psi_{\text{opt}}\, log\big(1 + 2Q_{\max}\big)$$

**Proof** For any agent $i$, any policy $\pi$, and any $(s, u)$, we have:

$$Q_i^\pi(s, u) = \mathbb{E}\left[\sum_{k=0}^{\infty} \gamma^k\, r_{t+k+1} \,\middle|\, s^t = s,\, u^t = u\right]$$

By triangle inequality and linearity of expectation, we get:

$$\left|Q_i^\pi(s, a)\right| \leq \mathbb{E}\left[\sum_{k=0}^{\infty} \gamma^k\, |r_{t+k+1}|\right]$$

$$\leq \sum_{k=0}^{\infty} \gamma^k\, r_{\max} = \frac{r_{\max}}{1 - \gamma}$$

Taking the supremum value over $(s, a)$ and over all policies $\pi$ gives:

$$\|Q_i^\pi\|_\infty \leq \frac{r_{\max}}{1 - \gamma} \quad \Longrightarrow \quad \|Q_i^*\|_\infty \leq \frac{r_{\max}}{1 - \gamma}$$

We define $Q_{\max} := \frac{r_{\max}}{1-\gamma}$. The same argument applies and results in:

$$\left|Q_i(\cdot,\cdot)\right| \leq Q_{\max} \quad \text{and} \quad \left|Q_{b_i}(\cdot,\cdot)\right| \leq Q_{\max}$$

By using the optimism signal:

$$\Psi_i^t := \psi_{\text{opt}} \left|Q_{b_i}(s_{b_i}^t, u_{b_i}^t) - Q_i(s_i^t, u_i^t)\right|, \qquad \psi_{\text{opt}} > 0,$$

and the reverse triangle inequality, we get:

$$\left|Q_{b_i}(s_{b_i}^t, u_{b_i}^t) - Q_i(s_i^t, u_i^t)\right| \leq \left|Q_{b_i}(s_{b_i}^t, u_{b_i}^t)\right| + \left|Q_i(s_i^t, u_i^t)\right|$$

$$\leq 2Q_{\max}$$

Therefore:

$$0 \leq \Psi_i^t \leq \psi_{\text{opt}}\, log\big(1 + 2Q_{\max}\big).$$

## B.3 Upper bound on policy divergence

**Definition B.3 (Maximum relative deviation)** *For state $s$, we define the maximum relative deviation between the policies of agent $i$ and its worst neighbor $w_i$ at time-step $t$:*

$$p_n^t(s) := \max_{u \in U: \pi_{w_i^t}^t(u|s) > 0} \left| \frac{\pi_i^t(u \mid s)}{\pi_{w_i^t}^t(u \mid s)} - 1 \right|$$

**Theorem B.4 (Upper bound of the repulsion)** *For fixed $\pi_{w_i^t}(\cdot|s)$, for state $s$, we define:*

$$p_n^t(s) := \max_{u \in U: \pi_{w_i^t}^t(u|s) > 0} \left| \frac{\pi_i^t(u \mid s)}{\pi_{w_i^t}^t(u \mid s)} - 1 \right|$$

*and we get for any state distribution $\mu$:*

$$
\begin{aligned}
\mathbb{E}_{s \sim \mu} \left[ D_{\mathrm{KL}}(\pi_i^t \| \pi_{w_i^t}^t) \right] &\leq \mathbb{E}_{s \sim \mu} \left[ \left( p_n^t(s) \right)^2 \right] \\
&= \mathcal{O}\left( \left( p_n^t \right)^2 \right) \\
&\leq c_1 \cdot \left( p_n^t \right)^2
\end{aligned}
$$

*with $c_1$ being a positive constant.*

**Proof.** Let $p_u := \pi_i^t(u \mid s)$ and $q_u := \pi_{w_i^t}^t(u \mid s)$, and set:

$$\kappa_u := \frac{p_u}{q_u} - 1$$

We have as a KL expansion:

$$
\begin{aligned}
D_{\mathrm{KL}}(p\|q) &= \sum_u p_u \log \frac{p_u}{q_u} \\
&= \sum_u q_u (1 + \kappa_u) \log(1 + \kappa_u)
\end{aligned}
$$

Using $\log(1 + x) \leq x$ for $x > -1$, we apply it to this equation:

$$
\begin{aligned}
D_{\mathrm{KL}}(p\|q) &\leq \sum_u q_u (1 + \kappa_u)\, \kappa_u \\
&= \underbrace{\sum_u q_u \kappa_u}_{= \sum_u (p_u - q_u) = 0} + \sum_u q_u \kappa_u^2 \\
&= \sum_u q_u \kappa_u^2
\end{aligned}
$$

Let $p_n^t(s) := \max_u |\kappa_u|$. Then, we obtain:

$$\sum_u q_u \kappa_u^2 \leq \left( \max_u |\kappa_u| \right)^2 \sum_u q_u = \mathcal{O}\left( p_n^t(s) \right)^2 \leq c_1 \cdot p_n^t(s)^2$$

Combining these equations, we obtain:

$$D_{\mathrm{KL}}(p\|q) \leq c_1 \cdot p_n^t(s)^2$$

**Corollary B.5 (Bound on the weighted pessimism signal)** *Let $\Gamma_i^t(s) := D_{\mathrm{KL}}\left( \pi_i^t(\cdot \mid s) \,\|\, \pi_{w_i^t}^t(\cdot \mid s) \right)$ and $0 \leq \beta^t \leq \beta_{\max}$. Then:*

$$0 \leq \mathbb{E}_{s \sim \mu}\left[ \beta^t \, \Gamma_i^t(s) \right] \leq \beta_{\max} \mathbb{E}_{s \sim \mu}\left[ \left( p_n^t(s) \right)^2 \right] = \beta_{\max} \mathcal{O}\left( \left( p_n^t \right)^2 \right)$$

**Theorem B.6 (Boundedness of Bayesian surprise)** *For state $s$, let $P = \tilde{\pi}_i^{t-1}(\cdot \mid s)$ and $R = \pi_i^t(\cdot \mid s)$. We assume $P(u) > 0$ for all available actions $u \in \mathcal{U}$. We define:*

$$q^t(s) \; := \; \max_{u:\, P(u)>0} \left| \frac{R(u)}{P(u)} - 1 \right|$$

*Then the Bayesian surprise satisfies:*

$$D_{\mathrm{KL}}\big(P \parallel (1 - \tau^t)P + \tau^t R\big) \; = \; \mathcal{O}\Big( \tfrac{\tau^{t2}}{1-\tau^t} \big(q^t(s)\big)^2 \Big)$$

$$\leq \; c_2 \cdot \tfrac{\tau^{t2}}{1-\tau^t} \big(q^t(s)\big)^2$$

*with $c_2$ being a positive constant.*

**Proof**  Let $Q = (1 - \tau^t)P + \tau^t R$. Using the KL–$\chi^2$ inequality for mixture distributions, we obtain:

$$D_{\mathrm{KL}}(P\|Q) \; \leq \; \frac{\tau^{t2}}{1 - \tau^t}\, \chi^2(R, P)$$

where $\chi^2(R, P) = \sum_u \frac{(R(u)-P(u))^2}{P(u)}$

Under the assumption that for all available actions $u$: $P(u) > 0$, we obtain:

$$\left| \frac{R(u)}{P(u)} - 1 \right| \; \leq \; q^t(s)$$

Hence:

$$\chi^2(R, P) = \sum_u P(u)\Big( \frac{R(u)}{P(u)} - 1 \Big)^2$$

$$\leq \; \big(q^t(s)\big)^2 \sum_a P(u)$$

$$= \mathcal{O}\big(q^t(s)\big)^2$$

Substituting this bound into the $D_{KL}$ first inequality results in:

$$D_{\mathrm{KL}}(P\|Q) \; \leq \; c_2 \cdot \frac{\tau^{t2}}{1 - \tau^t} \big(q^t(s)\big)^2$$

# C   BEMAS ALGORITHM

---

**Algorithm 1:** Balancing Extremes in Multi-Agent Systems

---

1 **Inputs:** Environment $\mathcal{E}$; agents $i \in \{1, \ldots, n\}$ with value head $Q_i(s, \cdot)$ and policy head $\pi_i(\cdot \mid s)$; replay buffer $\mathcal{B}$; horizon $T_{\max}$; exploration horizon $T_{\exp}$; $\epsilon$-greedy schedule $\epsilon^t$; EMA rate $\lambda \in (0, 1)$; Dirichlet forgetting $\varphi \in (0, 1)$ and weight $\omega^t > 0$; optimism scale $\psi_{\text{opt}} > 0$; shaping weights $\alpha^t, \beta^t \geq 0$.

2 **Initialize:** Target networks $\bar{Q}_i$; curiosity scores $c_0(i) \leftarrow 0$, reliability scores $f_0(i) \leftarrow 1$; Dirichlet beliefs $\boldsymbol{\xi}_i^0 \in \mathbb{R}_+^{|\mathcal{U}_i|}$ (all-ones).

    **for** $t = 0, 1, \ldots, T_{\max} - 1$ **do**

3

        **Observe & act**: Each agent $i$ receives $s_i^t$ and selects $u_i^t$ by $\epsilon^t$-greedy from $Q_i(s_i^t, \cdot)$.

4         **Environment step**: Receive $(s^{t+1}$ and the reward from the environment $r^t$.

5         **Phase-based neighbor scoring**, Calculate agents' *Phase score:*

$$\eta^t(j) = \begin{cases} c^t(j), & t < T_{\exp} \\ f^t(j), & t \geq T_{\exp} \end{cases}$$

        **for** *agent* $i \in \mathcal{V}$ **do**

6

            **Select best and worst neighbors:**

$$b_i^t \in \arg\max_{j \in \mathcal{N}_i^t} \eta^t(j), \qquad w_i^t \in \arg\min_{j \in \mathcal{N}_i^t} \eta^t(j)$$

7             **Obtain** $\pi_i^t(\cdot \mid s_i^t)$ and $\pi_{w_i^t}^t(\cdot \mid s_{w_i^t}^t)$ :

$$\Gamma_i^t = D_{\text{KL}}\big(\pi_i^t(\cdot \mid s_i^t) \,\big\|\, \pi_{w_i^t}^t(\cdot \mid s_{w_i^t}^t)\big)$$

8             **Query** $Q_{b_i^t}(o_{b_i^t}^t, a_{b_i^t}^t)$ from the best neighbor:

$$\Psi_i^t = \psi_{\text{opt}}\, g\left(\Delta_i^t\right) \mathbf{1}\{\Delta_i^t > 0\},$$

9             **Calculate Bayesian surprise**:

$$\Lambda_i^t = D_{\text{KL}}\big(\tilde{\pi}_i^t \,\big\|\, \tilde{\pi}_i^{t+}\big)$$

10             **Shape reward**:
$$\tilde{r}_i^t = \alpha^t\, \Psi_i^t + \beta^t\, \Gamma_i^t - \lambda^t\, \Lambda_i^t$$

        Push $\left(s^t, u^t, \tilde{r}^t, s^{t+1}, \text{done}\right)$ into $\mathcal{B}$.

        **if** $t$ *is an update step* **then**

11

            **Sample a minibatch** $\{(s, u, r, s', \tilde{r})\} \subset \mathcal{B}$.

12             **Get Shaped target:**

$$y_i = \tilde{r}_i + \gamma \max_{u'} \bar{Q}_i(s', u')$$

13             **minimize** $\left(y_i - Q_i(s, u)\right)^2$ and periodically **update** target network $\bar{Q}_i$.

        **if** done **then**

14

            reset episode.

---

## D EXPERIMENTAL DETAILS

### D.1 RESULTS

Table 2 reports the final performance of the three frameworks across different settings: Setting A: 20 predators in a 12 by 12 grid, Setting B: 20 predators in a 13 by 13 grid, and Setting C: 26 predators in a 12 by 12 grid.

For each method, we compute the mean reward over the last 10% of training steps across seeds. We additionally report variability measures (standard deviation, quartiles) to capture consistency in performance.

Table 2: Final performance across seeds (mean of last 10% of reward).

| Setting | Framework | Mean | Std | Min | Q1 | Median | Q3 | Max |
|---|---|---|---|---|---|---|---|---|
| | IQL | 8903.536 | **901.697** | 7346.8 | 8314.583 | 8908.320 | 9552.737 | 11027.040 |
| Setting A | PED-DQN | 16239.342 | 1075.178 | 14121.4 | 15288.440 | 16476.220 | 16866.673 | 18469.987 |
| | BEMAS | 20696.530 | 1380.958 | 17698.0 | 19779.210 | 20143.747 | 22375.327 | **23005.427** |
| | IQL | 9176.154 | 1692.834 | 6729.227 | 7780.487 | 8821.020 | 10612.067 | 12185.387 |
| Setting B | PED-DQN | 13191.881 | 844.722 | 11704.653 | 12621.837 | 13205.953 | 13763.447 | 14654.680 |
| | BEMAS | 16736.963 | **702.895** | 15726.520 | 16173.253 | 16514.153 | 17421.980 | **18142.747** |
| | IQL | 17999.944 | 1216.947 | 16555.089 | 17004.221 | 17474.827 | 19271.172 | 20244.429 |
| Setting C | PED-DQN | 40473.494 | 4168.849 | 32509.326 | 35551.655 | 42636.553 | 43398.694 | **45904.114** |
| | BEMAS | 37125.758 | **2006.306** | 33570.801 | 34859.250 | 37651.824 | 38989.972 | 39896.219 |

### D.2 ABLATION RESULTS

We present the results of the ablation study (see 5.2) in Table 3, where each framework variant isolates one of the two shaping terms—optimism or pessimism. The evaluation protocol is identical to that used above.

Table 3: Final performance across seeds (mean of last 10% of reward).

| Framework | Mean | Std | Min | Q1 | Median | Q3 | Max |
|---|---|---|---|---|---|---|---|
| $r + \beta\Gamma$ | 20805.351 | 1262.095 | 19298.520 | 19803.477 | 20192.833 | 21880.835 | **22268.658** |
| $r + \alpha\Psi$ | 18956.702 | 688.856 | 16609.333 | 18024.520 | 18355.073 | 18730.463 | 19597.413 |
| $r + \beta\Gamma - \Lambda$ | 20287.399 | **430.485** | 19415.680 | 20016.793 | 20266.740 | 20491.320 | 21374.680 |
| $r + \alpha\Psi - \Lambda$ | 18147.964 | 485.444 | 17084.573 | 17907.887 | 18154.980 | 18460.967 | 19498.107 |

## D.3 HYPERPARAMETERS

Below are the hyperparameters used across baselines and the proposed BEMAS framework. N/A indicates the item is not used by that method.

Table 4: Hyperparameters used in training.

| Hyperparameter | IQL | PED-DQN | BEMAS |
|---|---|---|---|
| Max Steps per Episode $T_{\mathrm{ep}}$ | 100 | 100 | 100 |
| Action Space $|\mathcal{U}|$ | 5 | 5 | 5 |
| Conv Filters / Kernel | 32 / 3×3 | 32 / 3×3 | 32 / 3×3 |
| MLP Hidden Layers | 64:64 | 64:64 | 64:64 |
| Discount Factor $\gamma$ | 0.99 | 0.99 | 0.99 |
| Learning Rate | $1 \times 10^{-4}$ | $1 \times 10^{-4}$ | $1 \times 10^{-4}$ |
| Optimizer | Adam | Adam | Adam |
| Minibatch Size $B$ | 32 | 32 | 32 |
| Replay Buffer Capacity | $2 \times 10^4$ | $2 \times 10^4$ | $2 \times 10^4$ |
| Target Update (steps) | 5000 | 5000 | 5000 |
| Test Interval (steps) | 4000 | 4000 | 4000 |
| Exploration $\epsilon_{\min}$ | 0.1 | 0.1 | 0.1 |
| Exploration Denominator $E$ | $5 \times 10^5$ | $5 \times 10^5$ | $5 \times 10^5$ |
| Curiosity Smoothing $\lambda$ | N/A | N/A | 0.6 |
| Performance Smoothing $\lambda$ | N/A | N/A | 0.6 |
| $\alpha_{\mathrm{init}}$ / $\alpha_{\mathrm{final}}$ | N/A | N/A | 1.0 / 0.1 |
| $\beta_{\mathrm{init}}$ / $\beta_{\mathrm{final}}$ | N/A | N/A | 0.1 / 1.0 |
| KL Epsilon $\epsilon_{kl}$ | N/A | N/A | $10^{-8}$ |
| Bayesian Stability Enabled | No | No | Yes |
| Stability Window $W_s$ | N/A | N/A | 10 |
| Adaptation Rate $\varphi$ | N/A | N/A | 0.95 |
| Optimism Parameter $\psi_{\mathrm{opt}}$ | N/A | N/A | 0.9 |

## LLM USAGE DISCLOSURE

Large language models were used in this work to aid with polishing the writing and presentation.

