# OpenReview forum: "Balancing Extremes: Exploiting the Performance Spectrum from Best to Worst in Multi-Agent Systems"
_ICLR.cc/2026/Conference — Submitted to ICLR 2026_

### Official Review · Reviewer_tkzG · 2025-10-29

**Soundness:** 3
**Presentation:** 3
**Contribution:** 3
**Rating:** 6
**Confidence:** 4

**Summary:**

The paper proposes BEMAS, a decentralized and proximity-aware reward-shaping framework for cooperative multi-agent reinforcement learning (MARL). The core idea is that agents can exploit the performance spectrum within their neighborhood: learning from the best peers (optimism) and avoiding the worst (pessimism). BEMAS includes: a peer-relative action-value gap bonus (optimism) encouraging exploration when outperforming the best neighbor; a KL-divergence–based repulsion (pessimism) penalizing similarity to the worst neighbor’s policy; a Bayesian stability regularizer (based on Dirichlet beliefs and Bayesian surprise) to stabilize updates; a phase-aware schedule that transitions from optimism to pessimism over training.

**Strengths:**

1. The idea of conbining optimism and pessimism for MARL is novel and intuitive, which I believe can emerge multi-agent cooperative behaviours.
2. The presetation of this paper is good and easy to understand, with clear equations and figures.
3. The design of BEMAS is sound, with all expected parts included.
4. Clear proofs of boundedness for each shaping term (logarithmic optimism, KL-based repulsion, Bayesian surprise) is rigor and seems correct.

**Weaknesses:**

My major concerns about this paper lies in the experiment part.
1. The number of baselines chosen is insufficient. I suggest the authors compare BEMAS with other decentralized methods and optimism and pessimism based methods, such as [1] and [2]
2. The experiments are only conducted on MPE environments with 3 scenarios. It is better to evaluate BEMAS on other partially observable environments.

Minor concerns
1. Is $T_{exp}$ a hyperparameter? If so, it is important to conduct ablation experiments on this parameter, since it control the influences of the two key parts of BEMAS.

[1] Conditionally Optimistic Exploration for Cooperative Deep Multi-Agent Reinforcement Learning, UAI 2023
[2] Tactical Optimism and Pessimism for Deep Reinforcement Learning, NeurIPS 2021

**Questions:**

See Weaknesses.

---

> ### Author Response · Authors · 2025-11-28
>
> We sincerely thank the reviewer for the constructive suggestions regarding baselines, experimental coverage, and parameter sensitivity. Your feedback motivated several additional experiments, including expanded MARL benchmarks and ablation studies. We are available to elaborate on any part of the revisions or provide more detail as needed.
>
> 1. Thank you for this insightful suggestion. Our primary focus in this work has
> been on MARL-specific shaping mechanisms, and we have now incorporated comparisons
> with other widely used decentralized and CTDE baselines (QMIX, VDN,
> MAPPO, and IPPO). We agree that adding optimism/pessimism-based methods such
> as COE [1] would further strengthen the evaluation. Due to current GPU allocation
> constraints, the COE experiments on Level-Based Foraging settings are still running, but they will be
> completed in the coming days and included in the revised version of the paper.
>
>
> 2. Thank you for this suggestion. We have extended our evaluation beyond cooperative MPE prey-predator by
> adding experiments on standard Level-Based Foraging benchmarks. We used the maps:
> 10x10-9p-4f which contains 9 players and 4 foods in a 10x10 grid, and 15x15-9p-1f, which contains 9 players and 1 food in a 10x10 grid, comparing BEMAS with widely used decentralized and
> CTDE baselines, including QMIX, VDN, MAPPO, and IPPO.
>
> Across these settings (Figures 1 and 2), BEMAS consistently matches or surpasses the
> value-based baselines (QMIX and VDN) and demonstrates competitive performance
> relative to CTDE methods. In the compact 10x10-9p-4f scenario,
> BEMAS converges similarly to MAPPO/IPPO, while in the more sparse or
> coordination-heavy 15x15 tasks, it maintains stable learning and
> achieves strong returns throughout training.
>
> These additional benchmark results and their analyses will be included in the
> revised version of the paper.
>
>
> - $T_{exp}$ ablation study:
>
> $T_{\exp}$ indeed acts as a hyperparameter controlling the relative influence
> of optimistic and pessimistic shaping. To study its effect, we performed two
> ablations in a 12x12 setting with 20 predators and 19 preys:
>
> Exploration horizon:
>
> As shown in Figure 3, varying $T_{\exp}$ to
> 250K or 750K reveals a clear trade-off: a short exploration phase accelerates
> early learning but converges to weaker final performance, while a prolonged
> exploration phase delays exploitation and similarly reduces final return.
>
> Phase-switch time:
>
> Figure 4 decouples $T_{\text{switch}}$ from
> $T_{\exp}$ by fixing $T_{\exp}=500$K and setting
> $T_{\text{switch}} \in \{250\text{K},\, 1.5\text{M}\}$. Switching too early
> prematurely emphasizes pessimism, suppressing exploration, while switching too
> late delays coordinated exploitation. The default synchronized schedule yields
> the best overall performance.
>
> Looking ahead, we plan to investigate automated mechanisms for
> selecting $T_{\text{switch}}$, which would enable
> task-dependent trade-offs between optimism and pessimism.
>
>
> Figure 1: https://github.com/Epsilon314159/BEMAS/blob/main/LBF_results/LBF-15x15-9p-1f.png
>
> Figure 2: https://github.com/Epsilon314159/BEMAS/blob/main/LBF_results/LBF_10x10-9p-4f.png
>
> Figure 3: https://github.com/Epsilon314159/BEMAS/blob/main/Ablations_results/Robustness_to_exploration.png
>
> Figure 4: https://github.com/Epsilon314159/BEMAS/blob/main/Ablations_results/Phase_switch_ablation.png
>
> [1] Conditionally Optimistic Exploration for Cooperative Deep Multi-Agent Reinforcement Learning, UAI 2023

---

### Official Review · Reviewer_ktSw · 2025-10-31

**Soundness:** 3
**Presentation:** 3
**Contribution:** 3
**Rating:** 6
**Confidence:** 4

**Summary:**

This paper proposes BEMAS, a decentralized MARL framework that exploits both best and worst performing peers during the training. The key idea is to balance optimism and pessimism through two reward-shaping signals. 1)  An optimism bonus that rewards agents when their Q values outperform those of their best neighbors, encouraging exploration. 2) A pessimism penalty based on the KL divergence from the worst-performing neighbors, discouraging imitation of poor strategies. and 3) A phase-based schedule controls the relative weight of optimism and pessimism across training, while a Bayesian stability regularizer mitigates instability from moving peer references. The approach remains fully decentralized during execution, relying only on local communication during training, which has potential scalablity towards larger number of agents. Empirical results on cooperative predator–prey grid-world benchmarks demonstrate that BEMAS improves both sample efficiency and training stability over IQL and PED-DQN baselines.

**Strengths:**

1. Originality: This paper introduces an interesting conceptual framing in MARL by explicitly leveraging both the best and worst performing peers through optimism and pessimism shaping. While optimism and pessimism have individually appeared in prior literature, their coupling within a decentralized, proximity-aware reward shaping framework is original. The phase-based scheduling and the integration of a Bayesian stability regularizer add further novelty and theoretical grounding.
2. Quality: The paper is technically solid, with clear mathematical formulations and boundedness proofs for all shaping terms. The authors provide rigorous theoretical analysis ensuring stability and bounded rewards, complemented by ablation studies that systematically isolate the effects of each component.
3. Clarity: The manuscript is well-written, logically organized, and supported by diagrams that effectively communicate the method’s mechanisms. Mathematical notation is consistent, and appendices provide comprehensive proofs and experimental details.
4. Significance: The work makes a potentially valuable contribution to the broader MARL community by proposing a general, decentralized shaping mechanism that enhances both exploration and stability without centralized training, which might inspire new directions in cooperative learning and stability regularization in distributed systems. While the experiments are somewhat narrow in current scope, the conceptual and methodological contributions are potentially significant and broadly applicable.

**Weaknesses:**

1. The evaluation is restricted to a single class of cooperative grid-world predator and prey environments, which are relatively simple and low-dimensional, which limits confidence in the generality and scalability of BEMAS to more complex or continuous MARL benchmarks. Testing on at least one higher-dimensional or partially observable domain would significantly strengthen the empirical claims.
2. The baselines used IQL and PED-DQN do not represent the state of the art in cooperative MARL in fully decentralized training manner. More recent decentralized or reward-shaping methods such as IPPO, I2Q, or influence-based exploration models) would provide a stronger and fairer comparison. Without these, the performance improvements have risk of overstating the practical advantage of BEMAS.
3. The optimism–pessimism transition is controlled by a manually designed, piecewise-linear schedule. The choice of hyperparameters appears ad hoc and is not empirically justified. A sensitivity analysis or an adaptive scheduling mechanism would help clarify whether performance is robust or heavily dependent on tuning.
4. This approach relies on bounded local message exchange during training, but the paper does not quantify the communication overhead or test robustness under limited or noisy communication.
5. The theoretical analysis focuses on boundedness and stability but does not address convergence properties or potential negative effects of non-stationary peer references. Explicitly discussing when the shaping might mislead learning would make the theoretical contribution more complete.

**Questions:**

Based on the weaknesses I described above, I have some questions for authors to disscuss.
1. Could the authors clarify how BEMAS would scale to larger or more complex environments, such as continuous control tasks or high-dimensional cooperative benchmarks (e.g., SMAC or MPE)?
2. Are there specific computational or communication bottlenecks that would arise in such settings? Providing either empirical results or scalability analysis could strengthen confidence in generalization.
3. Could the authors provide the performance of IPPO and I2Q algorithms on predator-prey scenario?
4. How would BEMAS behave if the communication radius ρ were smaller or dynamic? Clarifying this could strengthen the decentralization claim.
5. The proofs establish boundedness and stability, but do not address convergence to optimal joint policies. Can the authors discuss whether BEMAS inherits or violates any known convergence properties from standard TD-learning under decentralized settings?
6. Could the optimistic and pessimistic signals ever create conflicting gradients or lead to oscillations? Empirical evidence or theoretical discussion would help.
7. Since the motivation draws from group learning and social dynamics, could the authors comment on whether the BEMAS framework could extend to competitive or mixed-motive environments, beyond purely cooperative ones?
8. Were all methods trained under equal computational budgets such as the same number of updates, training steps, and network capacity? Clarifying fairness of comparison would be helpful.

I look forward to the discussion during the rebuttal period and hope the authors can provide further clarification and evidence to address the raised concerns.

---

> ### Author Response · Authors · 2025-11-28
>
> We warmly thank the reviewer for the insightful and encouraging review.
> Your questions on scalability, robustness, convergence, and mixed-motive extensions contributed greatly to refining the paper.
> We have addressed each point in detail and look forward to your thoughts and follow-up during the discussion period.
> We remain available to provide further clarification or additional results.
>
> 1. BEMAS scales naturally to larger or more complex environments because each
> agent compares itself to only a small size set of local peers within a fixed radius. This keeps computation independent of global agent density and allows the mechanism
> to operate efficiently even as the environment or population grows.
>
> To test robustness across different spatial resolutions, we reduced the grid size and evaluated BEMAS in
> 7x7 and 9x9 predator-prey maps with 20 predators and 19 preys. As shown in
> Figures 1 and 2, BEMAS maintains stable learning and yields good final returns, indicating that its shaping mechanism does not degrade in more
> compact or dynamic state spaces.
>
>
>
> The extension to continuous-control domains is conceptually straightforward,
> since the shaping terms operate on policies and value estimates. Exploring these settings is an important direction for
> future work.
>
>
>
> 2. Due to the limited duration of the rebuttal period, we provide a scalability
> analysis rather than full large-scale experiments. In BEMAS, neither
> communication nor computation becomes a bottleneck as environments grow. Each
> agent operates within its own partially observable space and selects only a
> small number of local peers when identifying its best and worst
> neighbors. This keeps both the communication load and the computational cost
> independent of the total number of agents or the global density of the system.
> Even if the communication radius increases, the peer-selection step can remain
> capped, ensuring that the complexity grows only with the local neighborhood
> size, not with the global environment.
>
>
> This design allows BEMAS to scale to larger or more complex
> multi-agent settings, and we include empirical scalability analysis in the
> revision to strengthen confidence in the generalization of the approach.
>
>
>
> 3. We thank the reviewer for the insightful suggestion of adding additional
> baselines and a standard MARL benchmark. While we originally treated the
> predator–prey task as a cooperative MPE setting, we have now incorporated the
> Level-Based Foraging (LBF) environment with different configurations (10x10-9p-4f, which contains 9 players and 4 foods in a 10x10 grid, and 15x15-9p-1f, which contains 9 players and 1 food in a 10x10 grid) and
> a broader set of baselines, including IPPO, MAPPO, QMIX, VDN, and COE. Across
> these LBF tasks, BEMAS performs competitively with strong CTDE methods and decentralized baselines.
>
>
> Regarding the predator–prey setting, we implemented I2Q (Figure 3) and found that it
> performed poorly in this environment, failing to achieve stable learning or
> meaningful returns. For IPPO, we include its performance on the LBF tasks (Figures 4 and 5),
> where it serves as a strong and informative decentralized baseline.
> Across the LBF settings, BEMAS consistently matches or surpasses the
> value-based baselines (QMIX and VDN) and demonstrates competitive performance
> relative to CTDE methods. In the compact 10x10-9p-4f scenario,
> BEMAS converges similarly to MAPPO/IPPO, while in the more sparse or
> coordination-heavy 15x15 tasks, it maintains stable learning and
> achieves strong returns throughout training.
>
>
>
> Figure 1: https://github.com/Epsilon314159/BEMAS/blob/main/Ablations_results/BEMAS_7x7.png
>
> Figure 2: https://github.com/Epsilon314159/BEMAS/blob/main/Ablations_results/BEMAS_9x9.png
>
> Figure 3: https://github.com/Epsilon314159/BEMAS/blob/main/Ablations_results/Baselines_coop_prey_pred.png
>
> Figure 4: https://github.com/Epsilon314159/BEMAS/blob/main/LBF_results/LBF-15x15-9p-1f.png
>
> Figure 5: https://github.com/Epsilon314159/BEMAS/blob/main/LBF_results/LBF_10x10-9p-4f.png

---

> > ### Author Response · Authors · 2025-11-28
> >
> > 4. We have not yet explored fully dynamic communication radius, as doing so may
> > require additional mechanisms to safely handle variable neighborhood sizes
> > without information loss. Conceptually, however, BEMAS can be extended to
> > this setting: at each timestep, an agent can simply rely on whichever
> > neighbors are currently available, and in the rare case where no neighbors are
> > present, the shaped reward reduces to the base reward plus the stability term,
> > ensuring a well-behaved update.
> >
> >
> > To evaluate sensitivity to smaller communication radius, we ran ablations where
> > each agent observes different restricted local region (communication radius within partially observable spaces of 4x4 and 10x10). As shown in
> > Figure 6, BEMAS remains stable and competitive even under
> > these communication constraints, indicating that it does not necessarily rely on
> > large neighborhoods.
> >
> >
> > 5. The boundedness and stability results established in the paper provide the
> > foundations needed to discuss convergence within the standard TD-learning
> > framework. Once the reshaped reward is shown to be uniformly bounded
> > (Theorems~4.2, 4.3, 4.4, and 4.6), the corresponding TD error is also bounded at every
> > timestep. Under bounded TD targets and learning rates that satisfy the classical
> > Robbins–Monro conditions, the TD update inherits the same asymptotic
> > convergence guarantees as value-based methods in decentralized settings.
> >
> > In addition, a Lyapunov-style argument can be constructed using the squared
> > Bellman error as a candidate Lyapunov function. Because the reshaped reward is
> > bounded, the temporal-difference update produces a Lyapunov derivative that is
> > non-positive whenever the Q-values deviate from their fixed point. This implies
> > that the update dynamics are contractive in expectation, and the learning
> > process remains stable over time rather than diverging or oscillating.
> >
> > Together, (i) bounded reshaped rewards, (ii) bounded TD errors,
> > (iii) Robbins–Monro learning-rate conditions, and (iv) a Lyapunov-decreasing
> > update rule ensure that BEMAS does not violate known convergence properties of
> > TD-learning in decentralized settings. Instead, it fits within the same
> > well-posed framework and converges under the standard assumptions for
> > value-based MARL.
> >
> > 6. In the BEMAS framework, the optimistic and pessimistic signals do not introduce
> > conflicting gradients in the usual sense, because BEMAS reshapes the reward
> > rather than modifying the policy or value gradients directly. The shaping terms
> > enter only through the scalar TD target, so their combined effect is additive
> > at the reward level rather than competing at the gradient level.
> >
> > Potential oscillations could in principle arise if the optimistic and
> > pessimistic signals changed abruptly over time; however, BEMAS is designed to
> > prevent this. First, both signals are provably bounded (Theorems~4.2 and~4.3),
> > which limits the magnitude of any update. Second, the Bayesian stability
> > regularizer (Sec.~4.3) penalizes sudden shifts in an agent’s policy via the
> > Dirichlet-based stabilizing term, ensuring smooth evolution of the policy
> > distribution.
> >
> >
> > 7. The current formulation of BEMAS is designed for cooperative MARL, where the
> > optimistic and pessimistic shaping terms align agents toward collectively
> > beneficial behavior. Extending the framework to competitive or mixed-motive
> > settings is possible in principle, but requires modifying how peer information
> > is interpreted. In such environments, “best” and “worst” neighbors would no
> > longer correspond to contributors to a shared goal, but to agents whose
> > policies provide strategic advantage or represent adversarial risk.
> >
> >
> > However, developing these extensions requires additional design choices around
> > game-theoretic incentives and partial alignment of objectives. Exploring how
> > optimistic and pessimistic shaping could guide behavior in mixed-motive or
> > competitive settings is an interesting direction for future work, and we
> > believe the local, decentralized nature of BEMAS makes it a promising starting
> > point for such scenarios.
> >
> >
> > 8. Yes, all methods were trained under comparable computational budgets. We used
> > the same number of environment steps, updates, learning rates, and training
> > protocols across all baselines, and network architectures were kept as close as
> > possible to their standard implementations. For a
> > detailed side-by-side list of hyperparameters used for IQL, PED-DQN, and
> > BEMAS, please refer to Appendix~D.3. Among the evaluated methods, IQL
> > uses the smallest network and therefore trains the fastest: in the
> > 20-predator / 19-prey 12x12 setting, IQL completes training in roughly
> > 6 hours, PED-DQN in about 10 hours, and BEMAS in approximately 14 hours. In
> > the LBF experiments, all baselines required around 5-6 hours of training.
> > These choices ensure a fair and balanced comparison across methods.
> >
> >
> > Figure 6: https://github.com/Epsilon314159/BEMAS/blob/main/Ablations_results/Communication_radius.png

---

### Official Review · Reviewer_b1sv · 2025-11-01

**Soundness:** 2
**Presentation:** 2
**Contribution:** 2
**Rating:** 2
**Confidence:** 5

**Summary:**

The paper proposes BEMAS, a decentralized multi-agent reinforcement learning framework that uses local communication during training to identify best and worst-performing neighbors. It introduces two reward-shaping signals: an "optimism" bonus for outperforming the best neighbor and a "pessimism" penalty for having a policy similar to the worst neighbor. A Bayesian stability regularizer is added to prevent policy divergence. The method is evaluated on a cooperative predator-prey grid-world and is shown to outperform IQL and PED-DQN baselines.

**Strengths:**

This paper studies an important problem in co-MARL. The idea of identifying the best and worst neighbor is novel.

**Weaknesses:**

1. Methdological Flaws: Both the curiosity and performance scores are derived from the TD-error magnitude. This creates a circular dependency: the quality of the shaping signal depends on the quality of the learned Q-function, which is itself being trained using the shaping signal. Therefore, the shaped reward can be unbounded and lead to unstable feedback loops.

2. Lack of Theoretical Understanding: The design of optimistic and pessimistic shaping term should be explained further, especially their interpretation when all agents are neighbors. To be specific for example, why we should penalize the similar behavior of one agent to another even though they are in different local states?

3. The empirical results seem inadequate, as they consider only one single simple environment.

4. The presentation can be improved. Figure 1 uses $a$ to represent agents instead of actions as in the main body. Section 4.4 should appear earlier. Equation 4 seems to be wrong.

**Questions:**

See weakness.

---

> ### Author Response · Authors · 2025-11-28
>
> We sincerely thank the reviewer for the careful reading and for raising important methodological and presentation-related concerns. Your comments helped us improve the clarity, correctness, and completeness of the theoretical and empirical analysis. We remain available to provide more details or additional analyses upon request.
>
> 1. We thank the reviewer for raising this point. While both the curiosity
> $c_t(j)$ and performance $f_t(j)$ scores are derived from TD-error magnitudes, they are computed as exponential moving averages of past
> TD errors and are used only to rank neighbors at each training step.
> They are empirical summary statistics instead of bootstrapped targets, and therefore
> introduce a strictly causal, one-directional dependency:  $Q(t) \rightarrow \delta(t) \rightarrow (c(t),f(t)) \rightarrow  \hat{r}(t) \rightarrow Q(t+1)$.
>
>
> The Q-function at step $t+1$ never depends on its own statistics, which
> prevents the algebraic feedback loop.
>
>
> BEMAS is trained off-policy, and every minibatch sampled from the
> replay buffer always contains the base environment reward. This means
> that the classical TD error $\delta(r)=r^{b}+\gamma \max_{u}Q(\cdot)-Q(\cdot)$
> is computed directly from the true environment signal, independent of any
> shaping. The shaping terms (optimism, pessimism, stability) are added only
> after sampling to form the reshaped reward $\hat r$, from which we
> compute a separate shaped TD error $\delta(\hat r)$ used for the Q-update. This separation prevents runaway amplification because all TD-error statistics ($c_t$ and $f_t$)
> are derived from past Q-values, not from the reshaped updates themselves.
>
>
> We provide explicit boundedness guarantees for each shaping term
> (Theorems 4.2, 4.3, 4.4, and 4.6), ensuring that the optimistic gap, behavioral repulsion,
> and Bayesian stability penalty remain finite for all time. These results imply
> that the modified Dec-POMDP admits well-posed returns and cannot diverge due
> to unbounded shaping.
>
> To address early-training instability directly, BEMAS includes a
> Bayesian stability regularizer (Sec.4.3) that penalizes abrupt policy shifts
> through bounded Bayesian surprise. This term explicitly controls
> the feedback arising from non-stationary peer references and prevents the type
> of positive-feedback escalation.
>
> We will clarify this temporal causality and include the boundedness discussion
> more prominently in the revised version of the paper.
>
>
>
> 2. We appreciate the reviewer’s request for a deeper explanation of the optimistic
> and pessimistic shaping terms, especially in cases where all agents are
> neighbors or occupy different local states.
>
> On optimistic shaping:
>
> The optimistic state-action gap is inspired by difference-reward principles:
> an agent compares the quality of its chosen state-action pair to that of its
> best-performing peer. Since policies are derived from $Q$-values (via
> Boltzmann sampling), this comparison captures behavioral quality rather
> than raw state similarity. The goal is not to imitate the neighbor’s exact
> state, but to propagate high-quality decision patterns under partial
> observability.
>
> On pessimistic shaping:
>
> The KL-based repulsion does not penalize similarity of raw observations; it
> penalizes similarity of policies, i.e., the action distributions that
> reflect each agent’s current estimate of good behavior. Even if two agents
> occupy different local states, their policies encode their learned strategic
> preferences. Avoiding the worst peer’s policy helps prevent convergence to
> suboptimal behavioral modes, which is particularly valuable in decentralized
> settings where miscoordination can arise from locally poor value estimates.
>
> When all agents are neighbors:
>
> If the communication radius is sufficiently large, agents effectively share a
> richer portion of the partially observable space. In this case, the
> proximity-based neighborhoods overlap, creating multi-hop information flow
> without requiring full observability or centralized critics. Penalizing
> similarity to the worst-performing policy in this expanded neighborhood
> still serves its intended purpose: discouraging the propagation of weak local
> strategies and mitigating collapse into poor equilibria.
>
> To illustrate this, Figure 1 includes an additional ablation where each agent observes
> a 10x10 region in the 12x12 map with 20 predators and 19 preys, making almost all agents
> neighbors; BEMAS still performs well under this dense-neighborhood setting.
>
> We will clarify these interpretations and their connection to partial
> observability and decentralized coordination in the revised version of the
> paper.
>
>
> Figure 1: https://github.com/Epsilon314159/BEMAS/blob/main/Ablations_results/Partial_observability_radius_10x10.png

---

> > ### Author Response · Authors · 2025-11-28
> >
> > 3. To address this concern, we have added results on standard Level-Based Foraging
> > benchmarks 10x10-9p-4f (contains 9 players and 4 foods in a 10x10 grid) and 15x15-9p-1f (contains 9 players and 1 food in a 10x10 grid) in Figures 2 and 3. We compare BEMAS against widely used CTDE and decentralized baselines
> > including QMIX, VDN, MAPPO, and IPPO.
> >
> > Across these settings, BEMAS consistently matches or surpasses the value-based
> > baselines (QMIX and VDN) and provides competitive performance relative to
> > strong CTDE methods. In the 10x10-9p-4f scenario,
> > BEMAS converges similarly to MAPPO/IPPO, while in the sparser or more
> > coordination-heavy tasks (15x15), it achieves steady learning
> > but still outperforms the strongest CTDE approaches in final return.
> >
> > We will include these additional benchmark results and their analyses in the revised version of the paper.
> >
> >
> > 4. We thank the reviewer for these helpful presentation-related comments. Figure 1 in the paper should indeed consistently distinguish agents from actions. In particular,
> > the TD-error equation should use $u$ for actions, while $a$ in the figure
> > correctly refers to agents. We will revise the notation throughout the paper to
> > ensure full consistency.
> >
> > Regarding Equation (4), the correct formulation of the reliability score is:
> > $f^{t}(j) = (1-\lambda) f^{t-1}(j) + \lambda \frac{1}{|\delta^{t}(j)|}$.
> > We appreciate the reviewer pointing this out and will correct it in the revised
> > version.
> >
> >
> > Finally, we agree that Section 4.4 would be clearer if introduced earlier. In
> > the revision, we will move it to the beginning of Section 4 and improve the
> > overall flow, notation, and presentation for clarity.
> >
> >
> >
> > Figure 2: https://github.com/Epsilon314159/BEMAS/blob/main/LBF_results/LBF-15x15-9p-1f.png
> >
> > Figure 3: https://github.com/Epsilon314159/BEMAS/blob/main/LBF_results/LBF_10x10-9p-4f.png

---

### Official Review · Reviewer_3HGc · 2025-11-03

**Soundness:** 3
**Presentation:** 3
**Contribution:** 3
**Rating:** 4
**Confidence:** 3

**Summary:**

This paper introduces BEMAS, a method that helps agents learn by looking at the best and worst performers in their local neighbourhood. It gives agents an optimistic bonus for outperforming their best neighbour (which encourages exploration) and a pessimistic penalty that encourages them to act differently than their worst neighbour (which helps them avoid bad strategies). The system is scheduled to focus on optimism early in training and then ramp up the pessimism later. A stability penalty is also added to prevent the agent's strategy from changing too suddenly. The authors prove these reward signals are stable. Experiments on a predator-prey game show BEMAS learns faster and achieves better or more consistent results than other methods. Further tests show that the pessimism signal, combined with the stability penalty, is the main reason for its strong performance late in training.

**Strengths:**

- Clear design that is easy to implement and reason about
- Fully local messaging during training; no comms at execution
- Boundedness results for each shaping term; shaped returns are well‑posed
- Good ablation narrative (optimism helps early & pessimism + surprise stabilise later)

**Weaknesses:**

- The optimistic “Q‑gap” compares different agents’ Q‑values in different states, which may be poorly calibrated and noisy
- TD‑error EMAs as curiosity/reliability proxies are not validated; could correlate weakly with true usefulness or competence
- Benchmarks/baselines are limited (single grid world; no QMIX/VDN/MAPPO etc.)
- Limited parameter sensitivity analysis ​
- No study of communication robustness

**Questions:**

- How do you calibrate cross‑agent Q‑values for the optimism gap? E.g. did you try advantage or z‑score normalisation?
- Any empirical correlation between $\delta$ and actual competence/return?
- Have you considered the sensitivity of results to various key parameters
- I'd like to see ablations on neighbourhood radius  and on message dropouts
- Can you add at least one standard MARL benchmark and baselines like QMIX/VDN or MAPPO?

---

> ### Author Response · Authors · 2025-11-28
>
> We sincerely thank the reviewer for the thorough and constructive feedback. We are encouraged by your positive assessment of the clarity, theoretical grounding, and ablation narrative of BEMAS. We have addressed each question in detail, and we are fully available to provide any additional clarification or further experiments if needed.
>
>
> - Thank you for this insightful question. We note that in our setting, all agents share identical reward scales, architectures, and discount factors, and the environment rewards are bounded. As a result, agents’ Q-values remain naturally comparable without explicit calibration. For example, at a representative training step, agents’ Q-vectors lie in similar ranges, e.g.: Agent 1: [0.055, –0.058, –0.053, –0.031, 0.013], Agent 5: [–0.052, 0.197, 0.166, 0.093, –0.123], Agent 2: [0.169,  0.008, -0.088, -0.293, -0.234], and Agent 18: [0.065, 0.325, –0.053, –0.068, –0.017].
>
> These values show that despite per-agent learning differences, magnitudes stay well aligned and within a narrow band, which makes the optimistic gap stable and meaningful.
>
> We implemented a normalization ablation in a 12x12 prey-predator setting with 20 predators and 19 preys. As shown in Figure 1, the normalized variant (“BEMAS_normalised”) learns smoothly but converges to lower final returns. Normalization may reduce the natural fluctuations in the Q-gap that carry important information about local competence, which weakens the optimistic exploration signal during training.
>
> - We computed the empirical Pearson correlation between the TD-error magnitude
> $\lvert \delta \rvert$ and the episodic return over the full 2M-step training horizon.
> Using the first, middle, and final third of training (0-0.67M, 0.67-1.33M,
> 1.33-2.0M steps), we observe a clear phase-dependent structure: in the early
> phase, $\lvert \delta \rvert$ is moderately negatively correlated with return
> ($-0.418$), reflecting that high TD-error still corresponds to unstable behavior.
> In the middle phase, the correlation becomes strongly positive ($+0.604$),
> indicating that higher TD-error reliably signals productive exploratory
> transitions. In the late phase, the correlation approaches zero ($-0.019$) as
> TD-errors flatten, and value estimates stabilize. These results confirm that
> $\lvert \delta \rvert$ carries phase-specific information, which
> supports our design of using opposite rankings in the exploration and
> exploitation phases.
>
>
> - We performed additional sensitivity studies on the schedule parameter: Phase-switch time in 12x12 setting with 20 predators and 19 preys.
> Figure 2 decouples $T_{\text{switch}}$ from $T_{\exp}$ by fixing
> $T_{\exp}=500$K and setting $T_{\text{switch}}\in\{250\text{K},1.5\text{M}\}$.
> Switching too early suppresses exploration; switching too late delays
> exploitation. The default synchronized choice performs best.
>
> Ablations isolating optimism and pessimism are provided in Section 5.2 of the
> paper. As future work, we plan to investigate automated schedules and
> task-adaptive weighting for all three shaping terms.
>
> - Thank you for the suggestion. We performed additional ablations on both
> (1) message dropout and (2) neighborhood radius in the 12$\times$12 setting with 20 predators and 19 preys.
>
> Message dropout:
>
> Figure 3 reports results under an extreme setting where each agent keeps only
> its four closest neighbors, but two of them are marked as unreliable and each
> of their messages is dropped independently with probability $0.5$. Even under
> this strong disruption, BEMAS remains relatively stable and continues to outperform the
> baselines, indicating that the shaping mechanism is robust to communication
> failures.
>
> Neighborhood radius:
>
> Figure 4 varies the size of the partially observable neighborhood from the default  7x7 to 4x4 and
> 10x10. The 4x4 setting yields slightly lower final returns due to
> more limited peer information, while the 10x10 setting shows higher variance and
> occasional instability because the rapidly changing large neighbor set makes the
> peer rankings less consistent across timesteps. Nonetheless, both variants remain close to
> the default BEMAS curve, indicating that the method is broadly robust across sparse and
> dense interaction regimes.
>
> Figure 1: https://github.com/Epsilon314159/BEMAS/blob/main/Ablations_results/Normalisation_ablation.png
>
> Figure 2: https://github.com/Epsilon314159/BEMAS/blob/main/Ablations_results/Phase_switch_ablation.png
>
> Figure 3: https://github.com/Epsilon314159/BEMAS/blob/main/Ablations_results/Message_dropout.png
>
> Figure 4: https://github.com/Epsilon314159/BEMAS/blob/main/Ablations_results/Communication_radius.png

---

> > ### Author Response · Authors · 2025-11-28
> >
> > - Thank you for this suggestion. We have added results on standard Level-Based Foraging
> > benchmarks 10x10-9p-4f (contains 9 players and 4 foods in a 10x10 grid) and 15x15-9p-1f (contains 9 players and 1 food in a 15x15 grid) in Figures 5 and 6. We compare BEMAS against widely used CTDE and decentralized baselines
> > including QMIX, VDN, MAPPO, and IPPO.
> >
> > Across these settings, BEMAS consistently matches or surpasses the value-based
> > baselines (QMIX and VDN) and provides competitive performance relative to
> > strong CTDE methods. In the 10x10-9p-4f scenario,
> > BEMAS converges similarly to MAPPO/IPPO, while in the sparser or more
> > coordination-heavy tasks (15x15), it achieves steady learning
> > but still outperforms the strongest CTDE approaches in final return.
> >
> > We will include these additional benchmark results and their analyses in the revised version of the paper.
> >
> >
> > Figure 5: https://github.com/Epsilon314159/BEMAS/blob/main/LBF_results/LBF-15x15-9p-1f.png
> >
> > Figure 6: https://github.com/Epsilon314159/BEMAS/blob/main/LBF_results/LBF_10x10-9p-4f.png

---

### Author Response · Authors · 2025-11-28

We sincerely thank all reviewers for their thoughtful and constructive feedback.

To support full reproducibility, we have added the code for the new Level-Based Foraging experiments and additional baselines to our GitHub repository: https://github.com/Epsilon314159/BEMAS/tree/main/Level_Based_Foraging .

These new experiments and results will be incorporated into the revised version of the paper, and we remain available to provide any further details as needed.

---

### Meta-Review · Area_Chair_edFU · 2026-01-05

**Summary:**

I will list the most important comments that the reviewers noted during the review process:
1) The optimistic “Q‑gap” compares different agents’ Q‑values in different states, which may be poorly calibrated and noisy
2) TD‑error EMAs as curiosity/reliability proxies are not validated; could correlate weakly with true usefulness or competence
3) Benchmarks/baselines are limited (single grid world; no QMIX/VDN/MAPPO etc.)
4) Limited parameter sensitivity analysis and no study of communication robustness.
5) The shaped reward can be unbounded and lead to unstable feedback loops.
6) The optimism–pessimism transition is controlled by a manually designed, piecewise-linear schedule.

**Reviewer Concerns:**

During the rebuttal, the authors did a good job of researching the various components of their model and in the parameter sensitivity analysis. The authors provided additional ablation studies for message dropout, neighborhood radius, a normalization ablation, and analyses for optimistic and pessimistic shaping terms.

However, some of the important remarks were not fully addressed:
1) Comparison: the authors compared BEMAS against widely used CTDE and decentralized baselines including QMIX, VDN, MAPPO, and IPPO, but the results provide only competitive performance.
2) Other benchmarks: the authors didn’t add any other results, and the current environment is very simple.
3) Manual optimism–pessimism transition: the authors agreed with this shortcoming.

Despite the additional baselines, the proposed method does not show significant superiority over them. The proposed approach has methodological drawbacks, including manual regulation of the optimism–pessimism transition.

**Reviewer Scores:**

1) Reviewer 3HGc (score 4) would most likely have left his initial score.
2) Reviewer b1sv (score 2) would most likely have left his initial score.
3) Reviewer ktSw (score 6) would most likely have left his initial score.
4) Reviewer tkzG (score 6) would most likely have left his initial score.

---

### Decision · Program_Chairs · 2026-01-26

Reject